

**Properties and mass transport differences across the Falkland Plateau between**
**1999 and 2010**
M. Dolores Pérez-Hernández (1,5), Alonso Hernández-Guerra (1), Isis Comas-
Rodríguez (1), Verónica M. Benítez-Barrios (1), Eugenio Fraile-Nuez (2), Josep L.
Pelegrí (3) and Alberto C. Naveira-Garabato (4)
(1) Instituto de Oceanografía y Cambio Global (IOCAG), Universidad de Las Palmas de Gran Canaria (ULPGC), Las
Palmas, Spain
(2) Centro Oceanográfico de Canarias, Instituto Español de Oceanografía, Santa Cruz de Tenerife, Spain
(3) Institut de Ciències del Mar, Consejo Superior de Investigaciones Científicas, Barcelona, Spain
(4) University of Southampton, National Oceanography Centre, Southampton, United Kingdom
(5) Department of Physical Oceanography, Woods Hole Oceanographic Institution, Woods Hole, USA
**Abstract**
Decadal differences in the Falkland Plateau are studied from full-depth
hydrographic data collected during the ALBATROSS (April 1999) and MOC2-Austral
(February 2010) cruises. Differences in the upper 100 dbar are due to changes in the
seasonal thermocline, as the ALBATROSS cruise took place in the austral fall while the
MOC-Austral in summer. The intermediate water masses seem to be very sensible to
the wind conditions existing on their formation area, showing cooling and freshening
for the decade as consequence of a higher Antarctic Intermediate Water (AAIW)
contribution and of a decrease of the Subantarctic Mode Water (SAMW) stratum. The
deeper layers do not exhibit any significant change in the water masses properties. The
Subantarctic Front (SAF) in 1999 is observed at 52.2-54.8ºW with a relative mass
transport of 32.6 Sv. In contrast, the SAF gets wider in 2010, stretching from 51.1 ºW to
57.2ºW (the Falkland Islands), and weakening to 17.9 Sv. Changes in the SAF are
mainly affecting the northward flow of Subantarctic Surface Water (SASW), SAMW
and AAIW/ Antarctic Surface Water (AASW). The Polar Front (PF) carries 24.9 Sv in
1999 (49.8-44.4ºW), while in 2010 (49.9-49.2ºW) it narrows and strengthens to 37.3 Sv.






## 1. INTRODUCTION

The Antarctic Circumpolar Current (ACC) flows eastwards around the Antarctic
continent, transporting between 100 and 150 Sv (1 Sv $\approx 10^6$ m$^3$ s$^{-1} \approx 10^9$ kg s$^{-1}$) [Orsi *et*
*al.*, 1995, Cunningham *et al.* 2003]. Along its path, it connects the Atlantic, Pacific and
Indian basins, exchanging heat and freshwater among other properties. Although
convergence of net fluxes estimates have been achieved on basin scales [Ganachaud and
Wunsch, 2003], the ACC flow into the Atlantic Ocean is critical to establish the
magnitude and pathways of the Southern Ocean contribution to the deep global ocean
ventilation.
Peterson and Whitworth [1989] suggested that the Subantarctic Front (SAF) and the
Polar Front (PF), where the major velocity bands of the ACC occur, turn northwestward
across the Falkland Plateau to the west of the Maurice Ewing Bank, along the
Patagonian continental slope. This was supported by Peterson [1992], who estimated
the large contribution of the ACC to the Falkland Current (60-70 Sv), revealing the
importance of the overflow of southern waters to the South Atlantic boundary
circulation. Peterson and Whitworth [1989] located the SAF near 53°W, as corroborated
by Arhan *et al.* [2002], at a location where the ocean depth is 2000 m. Several studies
have later examined the path of the PF around the Maurice Ewing Bank [Trathan *et al.*,
2000] and its branching around 49-50°W [Arhan *et al.*, 2002], with a possible
meandering of the front according to Naveira Garabato *et al.* [2002].
The first hydrographic cruise along the Falkland Plateau was carried out in 1999.
The ALBATROSS (Antarctic Large-scale Box Analysis and the Role of the Scotia Sea)
cruise explored the ACC through the Drake Passage and the Scotia Sea (Figure 1). The





data of this cruise has been used to estimate relative transport, water masses, fluxes and
mixing across the Plateau [Naveira-Garabato *et al*., 2003] and to provide with a detailed
explanation of the deep waters in the Scotia Sea [Naveira-Garabato *et al.*, 2002]. Later
on, this section has been confronted with hydrographic cruises carried out north and
south of the Falkland Plateau to achieve a better knowledge of this area [Arhan *et al,*
2002; Smith *et al.*, 2010.].

In this study, the water masses, relative geostrophic velocities and transports across

an almost zonal hydrographic section carried out in 2010 along the Falkland Plateau are
evaluated. These results are compared with those obtained from the 1999 cruise in the
same area [Naveira-Garabato *et al.*, 2003], with the objective of assessing possible
relative transport and water mass differences between the two realizations.

The paper is organized as follows. Section 'Data and Methods' presents the cruise,

data and methodology used in this study. Section 'Results' gives the description of the
different water masses existing on the study region, it shows the changes observed in
the $\theta/S$ isobaric surfaces, the location of the fronts, the results from Bindoff and
McDougall [1994] model and the changes in the relative geostrophic transport. This
paper concludes with a 'Discussion and conclusions' section that confronts our
estimations with the existing and provides with the concluding remarks of the research.

**2. DATA AND METHODS**

The MOC2-Austral cruise was carried out between February 8 and March 10, 2010,

on board the BIO Hespérides. As shown in Figure 1a, 27 full-depth CTD stations were
occupied across the Falkland Plateau, tracking along the casts previously conducted in
between 41ºW and 57ºW at the nominal latitude of 51°S during the ALBATROSS
cruise in April 1999 [Naveira Garabato *et al.*, 2003]. With a spatial separation of 30 to



50 km, temperature and salinity profiles were obtained using a SeaBird 911+ CTD with
dual conductivity and temperature sensors. The temperature sensor has an accurary of
0.001ºC. The conductivity sensors were calibrated on board with bottle sample
salinities. To that end, water samples were analyzed on a Guildline AUTOSAL 8400B
salinometer with accuracy better than 0.002 for single samples (salinity is expressed in
the Practical Salinity Scale).

Relative geostrophic velocities are estimated using the sea bottom as level of no-

motion. The water column is divided into 18 neutral density layers following the work
of Naveira Garabato et al. [2003], with modifications attending to the present water
masses (see Table 1). Weddell Sea Deep Water (WSDW) is not found along the
Plateau, thus its density layers are not considered here.

Bindoff and McDougall [1994] describe a model to evaluate the temperature and

salinity variations in the water column. This model relates the temperature and salinity
in both pressure and density changes through the following equation:

$$\left.\frac{d\psi}{dt}\right|_z = \left.\frac{d\psi}{dt}\right|_{\gamma^n} - \left.\frac{dp}{dt}\right|_{\gamma^n}\frac{d\psi}{dp}$$

which shows that for a given property ($\psi$, temperature or salinity), the variations along
isobaric levels ($\left.\frac{d\psi}{dt}\right|_z$) can be described as the sum of changes along isoneutrals surfaces
($\left.\frac{d\psi}{dt}\right|_{\gamma^n}$) and changes due to vertical displacements of the density surfaces, referred to as
heaving ($\left.\frac{dp}{dt}\right|_{\gamma^n}\frac{d\psi}{dp}$). This allows the comparison between the isobaric changes and the
sum of the two decomposed components, which represent the variations of the water
masses (warming and freshening) and the heaving. To apply this methodology,
temperature and salinity are interpolated onto a grid with a pressure interval of 20 db
(from 10 to 3500 db) and the following neutral density (kg m$^{-3}$) values: from 26 to 27.6



each 0.02, from 27.7 to 28 each 0.01 and from 28.005   to 28.5 each 0.005. This vector
is selected to properly represent the different structures found in the water column.

In addition, Sea surface height (SSH) was downloaded from AVISO

(http://www.aviso.oceanobs.com/, August 20, 2015) between February 10 and 20, 2010
for the MOC-Austral cruise.

**3. RESULTS**
**3.1 Water Masses**

Water masses in the study region are labeled following Naveira Garabato *et al.*

[2003]. The isoneutrals 26.90, 27.20, 27.60 and 28.00 kg m$^{-3}$ (red solid lines in Figure
1b) divide the water column into Subantarctic Surface Water (SASW), Subantarctic
Mode Water (SAMW), Antarctic Intermediate Water (AAIW) mixed with Antarctic
Surface Water (AASW), Upper Circumpolar Deep Water (UCDW) and Lower
Circumpolar Deep Water (LCDW), respectively.

Figure 2 shows that, in both cruises, the Circumpolar Deep Water (CDW) is the

most homogenous water mass. The UCDW in the ALBATROSS cruise presents a wider
temperature range and it is less homogeneous than in the MOC-Austral cruise. Figure 2
also exhibits that the stratums of AAIW+AASW and SAMW are quite different
between cruises. The AAIW+AASW stratum of the MOC-Austral cruise presents a
minimum that consists on temperatures below 1.2ºC and salinities around 34. This
minimum indicates that the contribution of AAIW is higher in 2010 than in 1999. In
contrast, in the same stratum, the ALBATROSS cruise shows a thicker layer of AASW.
The SASW in 1999 reaches higher salinities and temperatures than in the MOC-Austral
cruise (this can be better observed in Figure 1b grey dots). It is also worth mentioning
the existing difference between the SAWM stratums of both cruises, as the one of the





ALBATROSS cruise has a wider range of salinities than the one of the MOC-Austral
(Figure 2). The upper layers are less comparable as the cruises took place in different
seasons, which implies different precipitation/evaporation and winds that will directly
affect the SASW stratum.

**3.2 Fronts**
In Figure 3, the prominent slope of the $\gamma^n$-surfaces together with the intensified
relative velocities, point out the presence of the SAF and PF near their historically
reported locations [Orsi *et al.*, 1995]. In 1999, a northward-flowing jet accompanies the
SAF, extending the front's influence from the surface down to approximately 1500 m
between 52.2 and 54.8°W (Figure 3a, stations 160 to 165). In contrast, the SAF is
displaced to the west in 2010, extending from the Falkland shore (57.2ºW) to 51.1ºW
(stations 5 to 17) and the horizontal density gradient and associated relative geostrophic
velocities are weaker (Figure 3b). Regarding the PF, its quasi-barotropic presence and
effect on the water column is most noticeable in 2010 (Figure 3b, stations 20 and 21),
when it intensifies, displaying the strongest flow to the north around 49.5°W. Figure 3a
shows how this front is weaker in 1999, when it extends approximately in between 44.4
and 49.8°W and no intense jets are triggered by its presence (stations 146 to 156).
It can be observed how these fronts are revealed by the slopping isoneutrals,
suffering significant changes between the two oceanographic cruises. Therefore, it is
important to determine which variations in potential temperature and salinity are due to
water masses changes and which are caused by the displacement of the $\gamma^n$-surfaces.

**3.3 The θ/S Isobaric Changes**





152  Figure 4 reveals that in the decade, the waters shallower than 50 db (roughly the

153 SASW stratum) exhibit a significant increase of temperature and salinity, being of 0.5ºC

154 and 0.12, respectively (Figure 4a to d). This surface increase is probably caused by the

155 fact that the area was sampled in very different seasons: while the MOC-Austral cruise

156 took place in the austral summer, the ALBATROSS cruise was carried out during the

157 austral fall.

158  In the waters immediately beneath (from 50 db to 500db), the intermediate stratums

159 of SAMW and AAIW+AASW present a decrease of temperature of 0.8ºC and 0.4ºC,

160 respectively (Figure 4a and b). In contrast, while salinity for the AAIW+AASW stratum

161 also decreases 0.01, the salinity of the SAMW increases 0.02 (Figure 4c and d). In these

162 intermediate stratums, at roughly the location of the fronts (between stations 5-17 and

163 20-21), a remarkable decrease in temperature can be seen (Figure 4a). In between

164 stations 9-12, where the SAF stands, the UCDW exhibits a remarkable increase in

165 salinity. The same is observed in the area of the PF, where an increase of salinity is

166 registered at the UCDW and AAIW/AASW stratums (Figure 4 c).

167  The UCDW and LCDW do not show any significant changes in temperature. The

168 UCDW increases 0.01 in salinity while the LDCW doesn't show any significant

169 difference in salinity (Figure 4c and d).

171 **3.4 Results of applying the Bindoff and McDougal [1994] analysis**

172  The temperature and salinity isobaric changes, their decomposition and the sum of the

173 two components are plotted in Figures 5a and 5b, respectively. Except for certain depth

174 ranges, the sum of the components (grey line) compares reasonably well with the

175 isobaric change (black line, $\theta_z$ and $S_z$) indicating that the decomposition has been

176 successfully performed. The few discrepancies observed will be analyzed at the end of





the section.

The surface and intermediate temperature and salinity variations are affected by both

mechanisms; changes along neutral surfaces ($\theta_n$ and $S_n$, blue lines) and changes due to
vertical displacement of the isoneutrals ($-N\theta_z$ and $-NS_z$, red lines) (Figure 5a and b). In
the SASW stratum (pressure<100db) an increase of 0.7ºC in temperature and 0.1 in
salinity per decade is observed in Figure 5a and b, respectively. This increase can also
be observable in Figure 5c. These increases come together with a temperature-driven
vertical displacement of the isoneutrals (Figure 5a red line). As the cruises took place on
different seasons the most plausible explanation for this shoaling is the different depths
of the seasonal thermocline, being shallower in summer (2010) than in fall (1999).

In contrast with the upper layer, the SAMW and AAIW/AASW stratums present a

decadal decrease of temperature (-0.6ºC) and salinity (-0.07) in between 100dbar and
500dbar (Figure 5a and b). These changes can also be observed in the average $\theta$/S-
diagram for the AAIW/AASW (Figure 5c). The SAMW and AAIW/ASW stratums
occupy the same depth range, but the AAIW/ASW water mass spans over a higher area
(Fig 3). Hence, the decomposition shown in Figure 5a and b is mainly showing the
behavior of the AAIW/AASW stratum and, therefore, it doesn't match with the increase
in salinity observed in Figure 5c for SAMW. On Figure 5c the lines linking points of
equal pressure for the SAMW and AAIW/AASW stratums are not parallel to the
isopycnals, indicating as well, displacement of the isoneutrals surfaces. This
displacement is a deepening of the isoneutrals, mainly driven by the salinity. At the
level of the UCDW no changes are observed (Figure 5 a to c). In contrast, the LCDW
stratum shows a deepening of the isoneutrals driven by both temperature and salinity,
although no changes along neutral surfaces is observed (Figure 5 a to c).

As seen in Figure 5, the sum of the components compares reasonably well to the



isobaric changes. However, a careful inspection reveals some discrepancies, which take
place between 52 and 57ºW and around 49.5ºW. These are the approximate locations of
the SAF and PF fronts. These gradients cause the vertical displacement of more than
200 db for some isoneutrals, invalidating at these specific locations, the linear
expansion used to derive the proposed decomposition model, as was also found in the
Gulf Stream by Arbic and Owens [2001]. Thus, in Figure 6 a sensitivity analysis is
carried out by using the model of Bindoff and McDougall [1994] without the stations
involved in the fronts, taking into account only the stations 18 (157) and 28 to 31 (145
to 142) for the 2010 (1999) survey. For the surface SASW water mass, the same
behavior is found with or without fronts: an increase of temperature and salinity, though
slightly higher in the decomposition done without the fronts (0.9ºC and 0.15 *vs.* 0.7ºC
and 0.1), and a temperature-driven shoaling of the isopycnals. Likewise, in the range
100-500dbar, where the SAMW and AAIW/AASW stratums appear, the decomposition
shows the same pattern; a slightly smaller decrease in temperature (-0.4ºC) and salinity
(-0.04) again accompanied with a salinity-driven deepening of the isopycnals.
In contrast, from 500 dbar to the bottom two differences appear between both
decompositions. The first one occurs in between 500 dbar and 2000 dbar, in both
decompositions a slightly increase of temperature and salinity is observed but in the one
carried out without the fronts it appears with a salinity-driven shoaling of the
isopycnals. This depth range is mainly occupied by the UCDW stratum. The second
significant change between both decompositions appears at the bottom of the profile, at
the domain of the LCDW. As the stations east of the MOC-Austral station 28
(ALBATROSS station 148) are shallower than 2400 dbar, this decomposition is mainly
showing the changes suffered at stations 18. This station is located between both fronts,
and shows a temperature-driven deepening of the isopycnals. The result of this stratum



227 can be neglected, as one station cannot be considered statistically significant to provide

228 representative results.


230 **3.5 Relative Geostrophic Transport changes**

231  Some significant differences are observed in the relative mass transport estimates

232 for 1999 and 2010 across the hydrographic line along the Falkland Plateau (Table 1).

233 The accumulated transports evidence the important role played by the SAF and PF on

234 the relative mass transport across the section during both realizations (Figure 7). During

235 the MOC-Austral cruise the SAF-associated jet is displaced westward and weakens 14.7

236 Sv as compared with the ALBATROSS observations (Figure 4 and table 1), affecting

237 mainly the relative transport of the SASW, SAMW and AAIW/AASW stratums (Figure

238 7). The relative net transport is 9.2 Sv greater during the ALBATROSS cruise as an

239 outcome of a more intense SAF. In contrast, the location of the PF remains unchanged

240 between both cruises but it strengthens up to 37.3 Sv during the MOC-Austral cruise

241 (*vs*. the 24.9 Sv registered in the ALBATROSS survey), affecting the relative transports

242 of all water strata. In 2010, immediately east of the PF, at 47.8ºW, a countercurrent

243 appears carrying -8.8 Sv to the south. Figure 8 shows the average SSH for the 2010

244 MOC-Austral cruise with the aim to understand the source of this counter-flow. In this

245 Figure, the PF flows to the north around station 20 and partially diverting southward at

246 station 23. This meandering of the PF has already been reported in previous studies

247 [Naveira Garabato *et al.*, 2002].

248  As shown in Table 1, SASW relative geostrophic transport in 2010 is 2.3 Sv, a

249 slightly lower value than the one of 1999. Similar behavior can be observed in most of

250 the remaining layers; SAMW, AAIW/AASW, and UCDW, being the surface and

251 intermediate stratums the ones with the highest decadal transport differences. This is



presumably due to a stronger SAF in 1999 (32.6 Sv) than in 2010 (17.9 Sv) (Table 1).
The LDCW stratum does not registers the SAF due to the bathymetry. Thus, the only
northward contribution to this stratum is done by the PF, which is stronger in 2010 than
in 1999 from the SAMW stratum to the bottom. The total transports of the PF are 24.9
Sv in 1999 *vs.* 37.3 Sv in 2010.
Figure 9 exhibits the vertical structure of the calculated mass transport in the
different layers, which define each water mass. The geostrophic transports in
ALBATROSS (1999) and MOC Austral (2010) hydrographic cruises behave likewise
across the water column. The transports from the surface to the UCDW stratums are
affected by a noticeable northward net transport decrease of 10.6 Sv from 1999 to 2010.
In contrast, the LCDW exhibits an increase of 1.4 Sv of the northward flow in 1999 and

2010.


4. **DISCUSSION AND CONCLUSIONS**
The decadal differences in the Falkland Plateau are studied from full-depth
hydrographic data collected during the ALBATROSS (April 1999) and MOC2-Austral
(February 2010) cruises. Water mass changes are explored in terms of changes along
neutral surfaces and changes due to vertical displacements of $\gamma^n$-surfaces, applying the
model proposed by Bindoff and McDougall [1994]. Variability in the SAF and PF
location and mass transport is inferred from relative geostrophic velocities estimated by
using the sea-bottom as the level of no-motion.
The SASW stratum presents a wider range of salinities and temperatures in 1999
than in 2010 as shown in the θ/S diagram. In spite of this, the θ/S isobaric changes show
an increase of surface temperatures and salinities matching the Bindoff and McDougall
[1994] model's result for changes along neutral surfaces. The model also exhibits





shoaling of the isopycnals. The most plausible source for these differences is the fact
that the hydrographic cruises took place in different seasons (ALBATROSS in austral
fall and MOC-Austral in austral summer). Hence the seasonal thermocline has probably
changed its depth due to the different seasonal heating and precipitation.

SAMW expands over a higher depth range and presents a wider range of salinities

in 1999 than in 2010 (Figures 2 and 4). In contrast, the $\theta$/S diagram and isobaric
changes for the AAIW/AASW stratum show a decrease in temperature and salinity in
2010 when the AAIW/AASW occupies a higher depth range (Figures 2 and 4). As both
Bindoff and McDougall [1994] model estimations (with and without frontal zones)
agree in that the changes in the intermediate stratums are due to the displacement of $\gamma^n$-
surfaces, some changes are likely to have occurred between 1999 and 2010 in the
Falkland Plateau. The Bindoff and McDougall [1994] model reveals a deepening of the
isoneutrals at these levels, where the AAIW/AASW stratum occupies a higher depth
range than SAMW. An explanation for changes in those stratums can be found in
Naveira Garabato *et al*. [2009]. Figure 10a shows the mean wind stress of the winters in
the period 1998 - 2010. This figure is analogous to Figure 10a of Naveira Garabato *et*
*al*. [2009]. In the climatological mean a continuous wind stress magnitude spreads west
from South America (Figure 10a). Figures 10b and 10c exhibits the previous winter
anomalies to the ALBATROSS and MOC-Austral cruises, respectively. These
anomalies look very different between themselves. Figure 10b shows a large eastward
(positive) wind stress anomaly in the Southern Pacific. Naveira Garabato *et al*. [2009]
suggest that this structure causes a shift in the SAMW formation area. This matches
with the changes observed in Figures 2 and 4, where the SAMW stratum area is
reduced. It also agrees with the isobaric changes reported, a decrease in temperature of
0.8ºC and an increase in salinity of 0.02 from 1999 to 2010.



Naveira Garabato *et al*. [2009] also reported that the 1998-wind stress anomaly
pattern shown in Figure 10b generates a shutdown of the AAIW formation. Due to this,
a minimum of temperatures (<1.2ºC) and salinities (ca. 34) can be observed only for the
MOC-Austral cruise in Figures 1b and 2b. The shutdown of the AAIW formation in
1998 is responsible of the observed changes from 1999 to 2010 at this stratum. Across
the decade, the AAIW/AASW stratum increases the spanning area at intermediate
layers and suffers a decrease of 0.6ºC in temperature and of 0.07 in salinity, which is
accompanied with a deepening of the isoneutrals. Wind-driven changes in the ACC
isobaric surfaces were also observed in Böning et al [2008], where a deepening of the
isopycnals 27.2 and 27.4 kg m$^{-3}$ is described. The reported decrease of 0.07 in salinity
agrees with the decadal trend of the ACC at 300-500 dbar observed in Böning et al.
[2008] shown in their Figure 4. In contrast, they find an increase of temperature at the
same layer, probably due to the contribution of other intermediate waters into the ACC.
The SAF and PF undergo some displacements and variations in intensity between
1999 and 2010. The SAF in 1999 is observed at 52.2-54.8ºW with a relative mass
transport of 32.6 Sv and, while it is wider in 2010, reaching the Falkland Islands, it
weakens to roughly half of the transport (17.9 Sv). The SAF is the main path for the
northward flow of SASW, SAMW and AAIW/AASW into the Atlantic Basin. The PF
also contributes to this northward flow, being important for the UCDW and LCDW.
The PF in 1999 is located in the longitudinal range 49.8-44.4ºW carrying 24.9 Sv, while
in 2010 it narrows, centering on 49.9-49ºW and strengthening to 37.3 Sv. The PF in
2010 carries the highest relative northward transport of the study area, but nearly 8 Sv
of it recirculate back southward as seen in the SSH image. This meandering of the PF
was also observed in Naveira Garabato *et al.*, [2002].





To conclude, a seasonal change of the thermocline is observed in the surface layer.
The intermediate water masses of the study area seem to be very sensible to the wind
conditions existing on their formation area. Hence in 2010 an increase (decrease) of the
AAIW/AASW (SAMW) stratum is observed together with a cooling, freshening and
deepening of the isopycnals at this level. The CDW layers do not exhibit any significant
change in the water masses properties, being the most homogenous water mass.
However the LCDW exhibited a temperature and salinity driven deepening of the
isopycnals from 1999 to 2010. The net transport is 9.2 Sv weaker in 2010 than in 1999.
Fronts change their width and strength between cruises, being the SAF/PF in 1999
thinner/wider and stronger/weaker than in 2010.

**Acknowledgments**
This study has been performed thanks to the MOC2 (CTM2008-06438-C02-02/MAR)
and Sevacan (CTM2013-48695), financed by the Spanish Government. The
ALBATROSS cruise was funded by a Natural Environment Research Council
grant (GR3/11654). This work was completed while M.D. Pérez-Hernández was a
Ph.D. student in the IOCAG Doctoral Programme in Oceanography and Global Change.
The authors would like to thank David Sosa, Rayco Alvarado and, all the scientific team
and crew on board the BIO Hespérides for their hard work at sea during the MOC-
Austral cruise.





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





**Tables:**

**Table 1.** SAF, PF and net geostrophic mass transport (Sv) per cruise and water mass.

The last row shows the net transport, while the last column indicates the transport

difference between cruises.

| | ALBATROSS (1999) | | | MOC-Austral (2010) | | | Difference (2010-1999) | | |
|---|---|---|---|---|---|---|---|---|---|
| | SAF | PF | Net | SAF | PF | Net | SAF | PF | Net |
| SASW $\gamma^n<26.90$ | 3.2 | 2.0 | 4.5 | 1.9 | 1.6 | 2.3 | -1.3 | -0.4 | -2.2 |
| SAMW $27.00<\gamma^n<27.20$ | 5.1 | 1.0 | 4.9 | 1.4 | 1.8 | 2.2 | -2.9 | 0.8 | -2.7 |
| AAIW / AASW $27.30<\gamma^n<27.60$ | 15.5 | 8.3 | 16.8 | 10.9 | 9.7 | 11.8 | -4.6 | 1.4 | -5.0 |
| UCDW $27.70<\gamma^n<28.00$ | 8.6 | 12.0 | 12.4 | 3.8 | 20.1 | 11.7 | -4.8 | 8.1 | -0.7 |
| LCDW $28.05<\gamma^n$ | 0.1 | 1.6 | 0.3 | 0.0 | 4.1 | 1.7 | -0.1 | 2.5 | 1.4 |
| Net | 32.6 | 24.9 | 38.9 | 17.9 | 37.3 | 29.7 | -14.7 | 12.4 | -9.2 |



**List of Figures**

**Figure 1.** a) Hydrographic stations carried out during ALBATROSS (1999, red dots)
and MOC-Austral (2010, black dots) cruises. b) θ-S diagram for both cruises. Red solid
lines represent the $\gamma^n$ values (26.90, 27.20, 27.60 and 28.00 kg m$^{-3}$) defining the
different water masses in the region.

**Figure 2.** A volumetric potential temperature-salinity diagram for the a) ALBATROSS
and b) MOC-Austral cruises. Red solid lines represent the $\gamma^n$ values (26.90, 27.20, 27.60
and 28.00 kg m$^{-3}$) defining the different water masses in the region. Dot size and color
indicates the logarithm of counts.

**Figure 3.** Geostrophic velocity (positive northward) relative to the bottom for a) the
ALBATROSS cruise and b) the MOC-Austral cruise. Black dashed lines mark zero
velocities. Thick black lines stand for the representative isoneutrals (26.90, 27.20, 27.60
and 28.00 kg m$^{-3}$) defining the water masses in the region. Station numbering and the
fronts (SAF and PF) location are displayed on top axis.

**Figure 4.** Vertical sections of potential temperature (a) and salinity (c) differences in
isobaric levels, for the decade (2010-1999). The lines superimposed over the vertical
sections (grey lines for the 1999 section and black lines for the 2010 section) stand for
the isoneutrals defining the different water masses in the region (26.90, 27.20, 27.60
and 28.00 kg m$^{-3}$). Station numbering and the fronts (SAF (gray1999, black 2010) and
PF) location are displayed on top axis. Side panels show the zonally averaged
differences of temperature (b) and salinity (d), respectively (solid black lines) together



with their 95% confidence interval based on a Student's t-test  (dashed grey lines).

**Figure 5** Isobaric changes from 1999 to 2010 ($\theta_Z$, black line) decomposed into changes
along neutral surfaces ($\theta_n$, $S_n$, blue line) and changes due to the vertical displacement of
isoneutrals (-$N\theta_Z$, -$NS_Z$ red line) for (a) potential temperature and (b) salinity. The grey
line shows the sum of both components. The lower panel (c) shows the average profile
of $\theta$/S for each cruise together with the densities that divide the water column into the
different water masses (26.90, 27.20, 27.60 and 28.00 kg m$^{-3}$; red lines) and the link in
between points of equal pressure (dashed blue lines).

**Figure 6** Comparison between the isobaric changes from 1999 to 2010 carried out with
the whole dataset of each year (a and b, same as Fig 5) and without the stations were the
fronts are located (c and d).   Isobaric changes from 1999 to 2010 ($\theta_Z$, black line)
decomposed into changes along neutral surfaces ($\theta_n$, $S_n$, blue line) and changes due to
the vertical displacement of isoneutrals (-$N\theta_Z$, -$NS_Z$ red line) for temperature (a and c)
and Salinity (b and d). The grey line shows the sum of both components.

**Figure 7.** East to west accumulated relative geostrophic mass transport, computed
across the ALBATROSS and MOC-Austral hydrographic sections. Station numbering
and the fronts (SAF and PF) location are displayed on top axis. Note the different
vertical scales.

**Figure 8.** AVISO Sea Surface Height (SSH) for the MOC-Austral cruise. Isolines have


a separation of 5 cm.
**Figure 9.** Relative geostrophic mass transport per layer across the ALBATROSS and
MOC-Austral sections.
**Figure 10.** Maps of NCEP-NCAR a) winter (July-September) mean wind stress in Pa
(arrows, color indicates magnitude) for the time period 1998-2010. b) Winter wind
stress anomaly (Pa) for year 1998. c) Same as b) but for 2009.












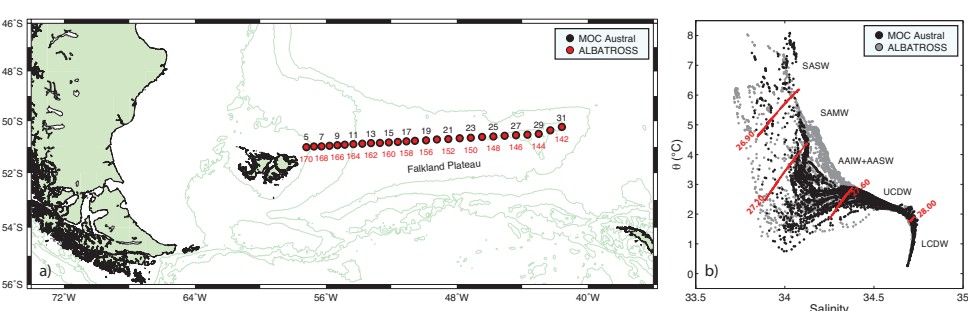


**Figure 1.** a) Hydrographic stations carried out during ALBATROSS (1999, red dots)

and MOC-Austral (2010, black dots) cruises. b) θ-S diagram for both cruises. Red solid

lines represent the $\gamma^n$ values (26.90, 27.20, 27.60 and 28.00 kg m$^{-3}$) defining the

different water masses in the region.




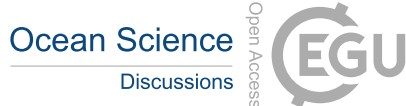







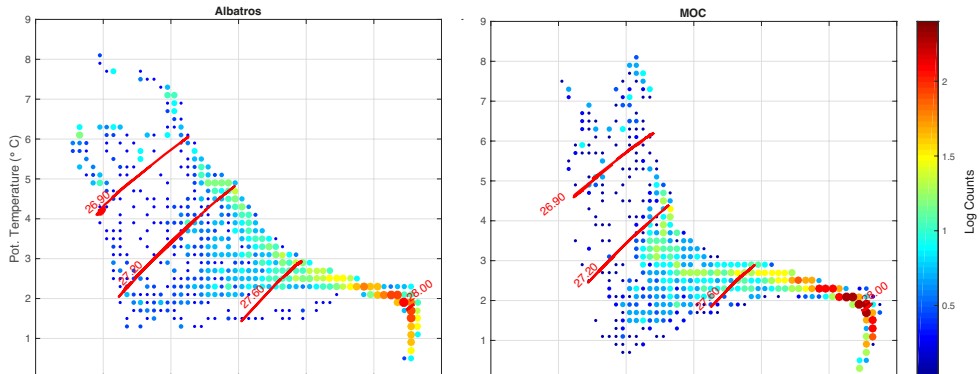


**Figure 2.** A volumetric potential temperature-salinity diagram for the a) ALBATROSS

and b) MOC-Austral cruises. Red solid lines represent the $\gamma^n$ values (26.90, 27.20, 27.60

and 28.00 kg m$^{-3}$) defining the different water masses in the region. Dot size and color

indicates the logarithm of counts.





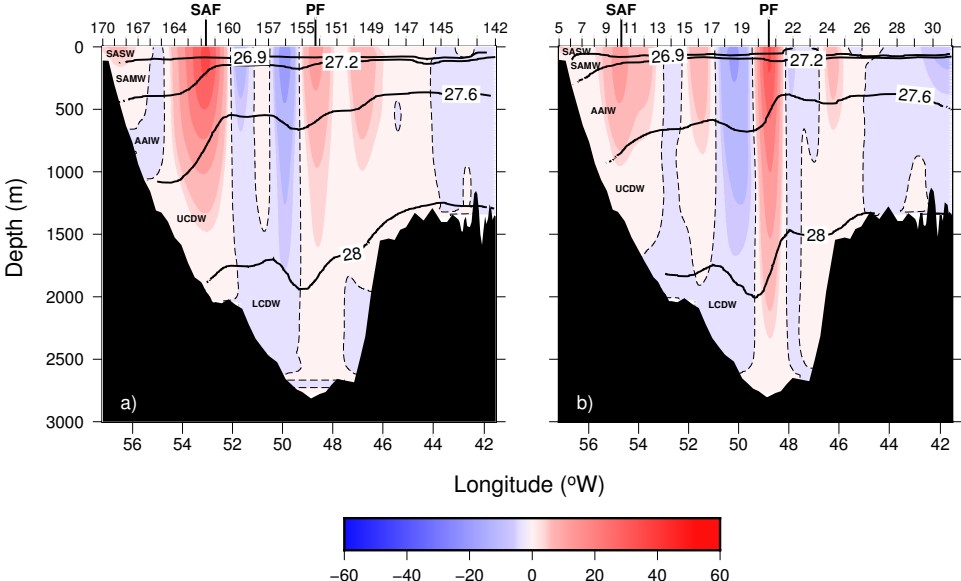


**Figure 3.** Geostrophic velocity (positive northward) relative to the bottom for a) the

ALBATROSS cruise and b) the MOC-Austral cruise. Black dashed lines mark zero

velocities. Thick black lines stand for the representative isoneutrals (26.90, 27.20, 27.60

and 28.00 kg m$^{-3}$) defining the water masses in the region. Station numbering and the

fronts (SAF and PF) location are displayed on top axis.

520

521




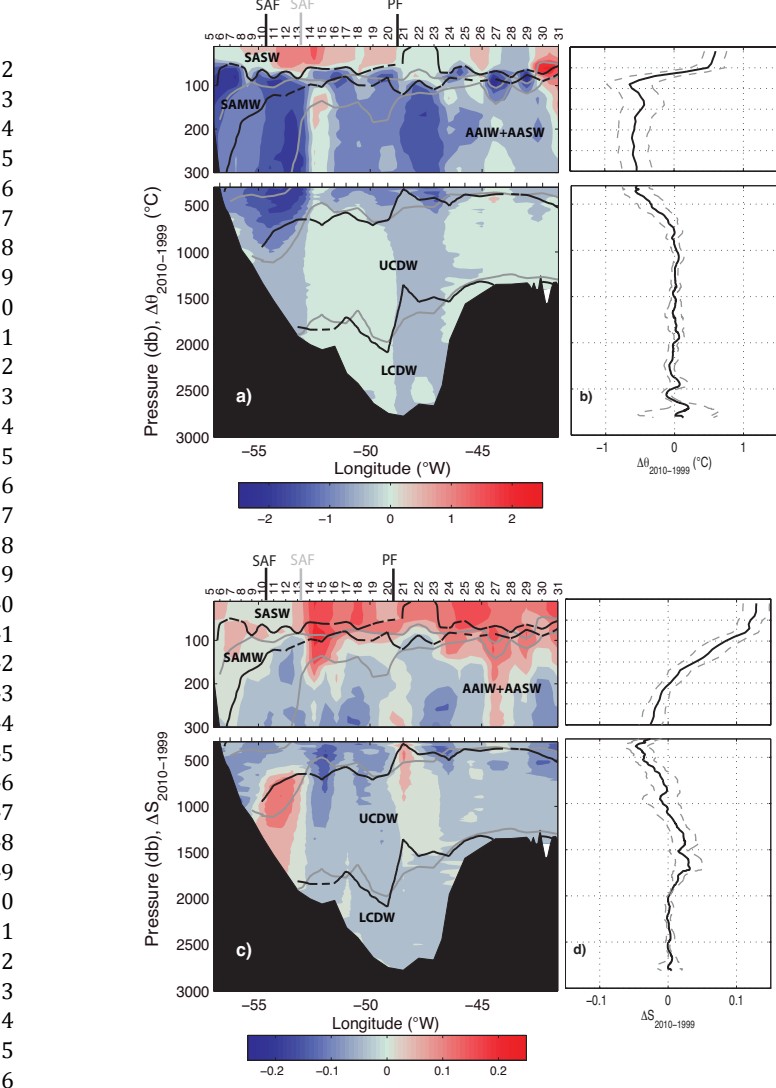

**Figure 4.** Vertical sections of potential temperature (a) and salinity (c) differences in isobaric levels, for the decade (2010-1999). The lines superimposed over the vertical sections (grey lines for the 1999 section and black lines for the 2010 section) stand for the isoneutrals defining the different water masses in the region (26.90, 27.20, 27.60 and 28.00 kg m$^{-3}$). Station numbering and the fronts (SAF (gray1999, black 2010) and PF) location are displayed on top axis. Side panels show the zonally averaged differences of temperature (b) and salinity (d), respectively (solid black lines) together with their 95% confidence interval based on a Student's t-test (dashed grey lines).



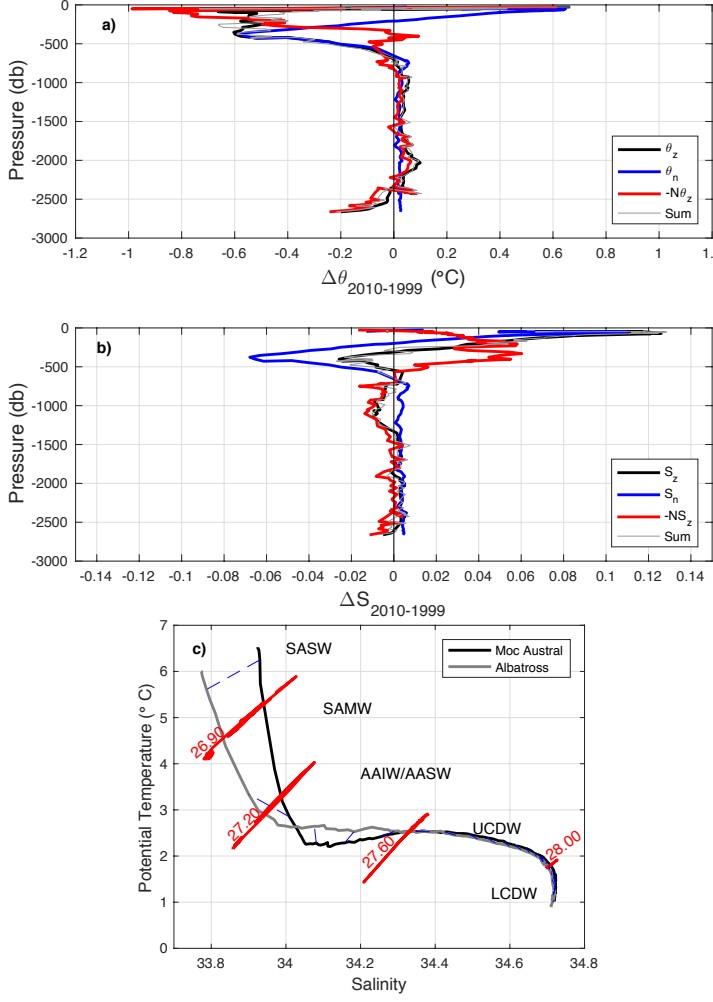

**Figure 5** Isobaric changes from 1999 to 2010 ($\theta_Z$, black line) decomposed into changes
along neutral surfaces ($\theta_n$, $S_n$, blue line) and changes due to the vertical displacement of
isoneutrals ($-N\theta_Z$, $-NS_Z$ red line) for (a) potential temperature and (b) salinity. The grey
line shows the sum of both components. The lower panel (c) shows the average profile
of $\theta/S$ for each cruise together with the densities that divide the water column into the
different water masses (26.90, 27.20, 27.60 and 28.00 kg m$^{-3}$; red lines) and the link in
between points of equal pressure (dashed blue lines).





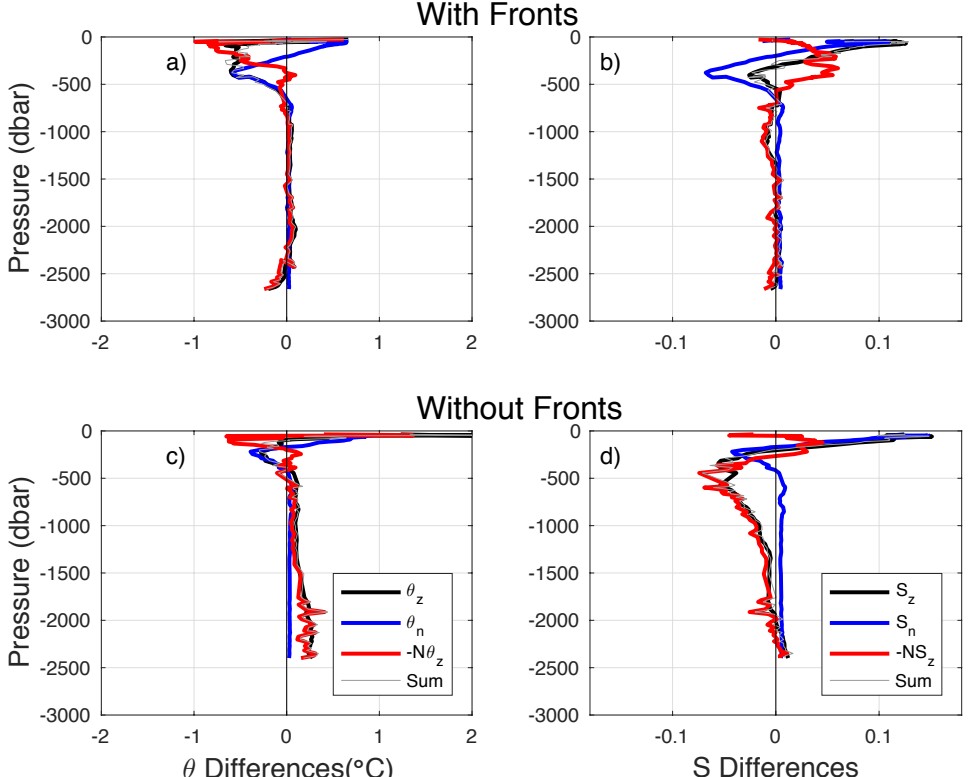


**Figure 6** Comparison between the isobaric changes from 1999 to 2010 carried out with

the whole dataset of each year (a and b, same as Fig 5) and without the stations were the

fronts are located (c and d). Isobaric changes from 1999 to 2010 ($\theta_z$, black line)

decomposed into changes along neutral surfaces ($\theta_n$, $S_n$, blue line) and changes due to

the vertical displacement of isoneutrals ($-N\theta_z$, $-NS_z$ red line) for temperature (a and c)

and Salinity (b and d). The grey line shows the sum of both components.

581
582





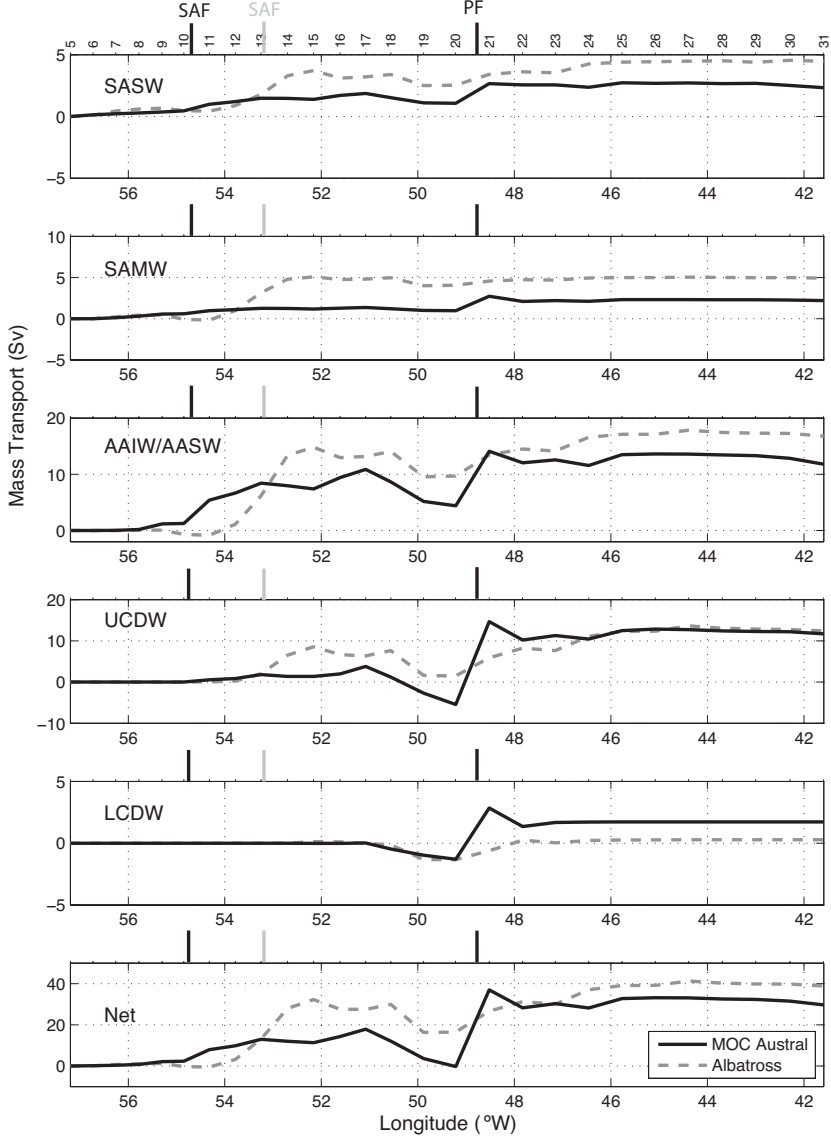

**Figure 7.** East to west accumulated relative geostrophic mass transport, computed across the ALBATROSS and MOC-Austral hydrographic sections. Station numbering and the fronts (SAF and PF) location are displayed on top axis. Note the different vertical scales.





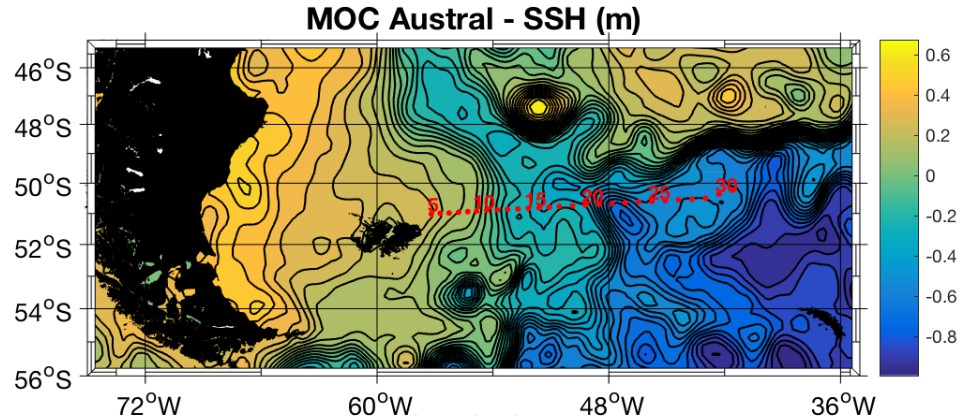

**Figure 8.** AVISO Sea Surface Height (SSH) for the MOC-Austral cruise. Isolines have

a separation of 5 cm.





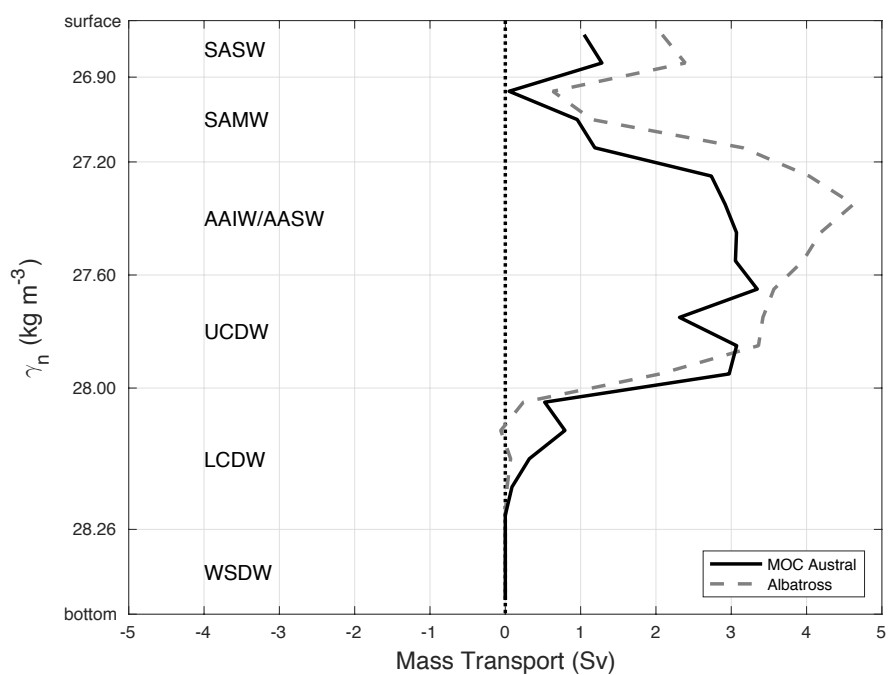


**Figure 9.** Relative geostrophic mass transport per layer across the ALBATROSS and

MOC-Austral sections.







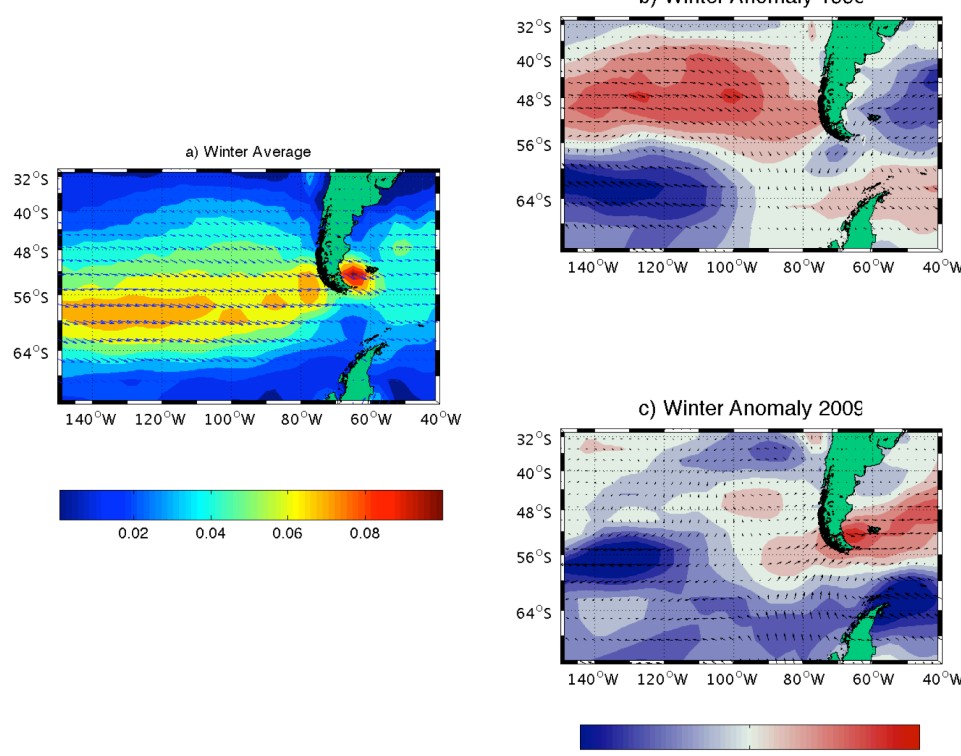


**Figure 10.** Maps of NCEP-NCAR a) winter (July-September) mean wind stress in Pa
(arrows, color indicates magnitude) for the time period 1998-2010. b) Winter wind
stress anomaly (Pa) for year 1998. c) Same as b) but for 2009.