# Peer review of "Differences between 1999 and 2010 across the Falkland Plateau: Fronts and water"

_Ocean Science, 2016_

## Referee Comment (RC1) · Anonymous Referee #1 · 5 Feb 2017

Review of the paper by M. Dolores Pérez-Hernández, Alonso Hernández-Guerra, Isis Comas-Rodríguez, Verónica M. Benítez-Barrios, Eugenio Fraile-Nuez, Josep L. Pelegrí, and Alberto C. Naveira-Garabato

"Properties and mass transport differences across the Falkland Plateau between 1999 and 2010"

The authors present a very detailed analysis of water masses in the region of the Falkland Plateau and their comparison between 1999 and 2010 on the basis of two hydrographic sections in the region. However, the main goal of their study is comparison between the properties and mass transport across the Falkland Plateau. The analysis of the differences and causes is not sufficient for immediate publication.

This region is the location of the beginning of the Falkland Current but the authors just mention this fact as not very important.

The cruises were made in different seasons of the year: the austral fall and summer (April and February). The changes in the water properties and dynamics between these two seasons can occur not only in the surface layer as the authors report. It is well known that the seasonal changes in the Falkland Current are very strong. The changes in the geostrophic component of the Falkland Current may reach the depths of 2000 m. Since the changes in the Falkland Current exist, similar changes may occur in the region where the current starts. Some analysis of what had happened between 1999 and 201 is needed. The authors cite a publication by BÅŚning et al. [2008], but this paper analyzes the changes in the ACC caused by decadal changes in the wind field. I would appreciate an analysis of the AVISO data in the region and variations in the geostrophic currents during the study period (or at least in the period when the AVISO data are available and reliable). The analysis of the seasonal changes in the geostrophic currents is important. Then, this analysis should be linked with the observations performed with an interval of 11 years. The changes that occur over a period of 11 years and the seasonal changes should be separated. The changes in mass transport even in the subsurface layers can be associated with the seasonal changes in winds. I am sure that some CTD data from the stations occupied in the region in the years between 1999 and 2010 can be found in the databases and added to the analysis. The main drawback of the analysis presented in this manuscript is complete absence of any data related to the period of 11 years.

I can recommend this manuscript for publication only if such analysis would be added. Major revision is currently needed.

A few minor remarks. Do not use different notations in one figure. Red and black dots in Fig. 1a and black and gray dots in Fig. 1b. Is Fig 1b is just a copy of the figure from another text?

What are the units of color scale in Figs. 3 and 4? I recommend changing color in one of the color scales not to use tones of red and blue in both color scales with different units. Do not use the same colors for different properties.

---

## Referee Comment (RC2) · Anonymous Referee #2 · 27 Mar 2017

General comments: This manuscript presents new data from 27 full-depth CTD stations across the Falkland Plateau in February/March 2010. The CTD stations were deliberately located at the same latitude and longitude positions of the April 1999 ALBATROSS cruise, to allow comparisons between the two data sets. The authors compare locations of fronts, water masses, and geostrophic volume transports (net, and for each water mass). This area is not an easy area to collect oceanographic data, because it is remote and has the Sub-Antarctic Front and Polar Front passing through it. The data presented are therefore important, and deserving of publication. Although well-written, the paper is lacking in some explanations, as detailed below in the specific comments. With revisions, this manuscript would be a useful addition to the literature.

Specific comments: As noted above, although the data are important and deserving of publication, there are some revisions needed.

Line 1: The title "Properties and mass transport differences across the Falkland Plateau between 1999 and 2010" does not express what the importance of the paper is. The values calculated are "volume transport" (units of metres cubed per second), not "mass transport" (units of kilograms per second). "Properties" is too vague – which properties? A title that summarises the key message or focus of the paper is needed.

Lines 13-28: The abstract summarises some of the main points, but does not tell the reader why they should read the full paper. At line 23, the term to "relative mass transport" is distracting, because what the authors have really looked at is "volume transport". The changes in width and volume transport of the SAF and PF from 1999 to 2010 are listed, but it is not clear why these have occurred or why the changes might be important. The abstract should be rewritten to draw the reader into the full paper.

Lines 34-36: The authors set the scene at the opening lines of their paper by citing Orsi et al. (1995) and Cunningham et al. (2003) in regards to ACC transport being between 100 and 150 Sv. However, there are a number of very recent papers that indicate that ACC transport might be much higher than this (Donohue et al. (2016) and references therein (at page 11,766)). It would be worth the authors including discussion of the implications (if there are any) of those papers to the research presented in this manuscript.

Lines 87-88: With respect to the 2010 data, the authors state: "Relative geostrophic velocities are estimated using the sea bottom as level of no-motion." However, Naveira Garabato et al. (2003) state for the 1999 data : "At each station pair, the reference velocity is initialized by performing a least squares fit of the geostrophic shear profile to the average detided water-track LADCP profile", which (if I am reading this correctly) means the 1999 data used a different method to obtain geostrophic velocities (by adjusting them to LADCP measurements). Naveira Garabato et al. (2003) in their section

3 then go on to discuss the issues with level of no motion assumptions. The authors of the present manuscript therefore need to explain why they used the level of no motion approach for the 2010 data, and whether they re-analysed the 1999 data using the same assumption or if they are using the 1999 data as processed by Naveira Garabato et al. (2003).

Table 1 (lines 425 onwards): The first three columns are ALBATROSS (1999) data, but the values in the table do not seem to match those given in Figure 14 of Naveira Garabato et al. (2003). For example, Table 1 states UCDW volume transport was 12.4 Sv, whereas Naveira Garabato give that volume transport as $23.5 \pm 4.5$ Sv. From reading the manuscript, I thought the same data was being used, so if a new analysis has been done, or there is some other reason for the values being different, then this should be explained.

Lines 265-335 (Discussion and conclusions): The discussions and conclusions do not really convey why the research was important. The main conclusions as presented in this section seem to be: (1) the observed changes in surface waters are because the measurements in 1999 (April=austral fall) were performed in a different season to 2010 (February=austral summer); (2) the intermediate waters underwent changes that are attributable to wind changes at their regions of formation; (3) the salinity and temperature do not change much for the CDW layers.

However, the conclusions above do not really connect with some of the statements in the "Results" section, e.g., lines 163-169, where it states "In between stations 9-12 where the SAF stands, the UCDW exhibits a remarkable increase in salinity." I also found it surprising that the authors did not directly mention the 1998-1999 ENSO and SAM events that Naveira Garabato et al. (2009) say were so important for the AAIW properties, and the implications therefore of comparing the ALBATROSS results with the 2010 ones. There needs to be some overall important conclusions presented from the research.

Technical corrections: Lines 3-5: It is completely up to the authors how they wish to have their names presented, but I have never seen Professor Naveira Garabato's paternal and maternal surnames hyphenated before.

Line 56: "data" is plural, so this line should read "data from this cruise have been..."

Line 57: Delete "with"

Line 59: "confronted" is the wrong word to use here.

Line 66: "realizations" is the wrong word to use here

Lines 67-73: This paragraph basically says that the paper is organized with data and methods first, then results, then discussions and conclusions. Almost all scientific papers are organised that way, so this paragraph could be deleted. A more useful thing to have in its place would be a sentence that starts with "In this paper,...", followed by the most important thing(s) that the paper shows/examines.

Line 82: "accurary" should be "accuracy" (presumably)

Line 83: An accuracy of $0.001°C$ would mean that the SBE911+ CTD on the ship was new from the factory or had been recently calibrated (or post-cruise factory calibrated), because the specifications of the SBE911+ CTD say this is the initial accuracy, with a drift of "$0.0002°C$ per month" (http://www.seabird.com/sbe911plus-ctd, under specifications tab). This may not be important for the level of accuracy required for the measurements presented in this manuscript, but more details should be given.

Line 86: I have no issues with the Practical Salinity Scale (rather than Absolute Salinity from TEOS-10) being used in this research, especially because the authors are making comparisons with earlier published work that was published using PSS. However, the reference for PSS should be cited at the end of this sentence.

Lines 89-90: "...modifications attending to the present water masses" – it is not clear to me what this means.

Line 193: "doesn't" is too informal for a scientific paper, please change to "does not".

Lines 235-236 and 238-239: At lines 235-236, the authors state "...the SAF-associated jet is displaced westward and weakens 14.7 Sv as compared with the ALBATROSS observations", then at lines 238-239 the authors state "The relative net transport is 9.2 Sv greater during the ALBATROSS cruise as an outcome of a more intense SAF." It would be easier for the reader to follow this line of argument if the comparisons were consistent, so something like this: "...the SAF-associated jet in 2010 was displaced westward and was weaker by 14.7 Sv compared with the ALBATROSS observations", then "The relative net transport is 9.2 Sv less during 2010 compared to during the ALBATROSS cruise as an outcome of a weaker SAF in 2010." How 14.7 Sv less volume transport of the SAF leads to 9.2 Sv reduction in net transport would also be worth explaining , that is why volume transport changes are not directly additive.

Lines 255-256: "The total transports..." – the authors have already stated these results at lines 239-241, so this sentence could be removed.

Figure 1 at lines 491-496: It looks like the 2010 stations were at the same locations as the ALBATROSS stations on the scale of this map, so rather than saying "red dots" and "black dots", perhaps just "red station numbers" and "black station numbers" would be sufficient. The figure caption should also say "Hydrographic stations across the Falkland Plateau...", because ALBATROSS included other stations not shown on this map.

Figure 4 at lines 522-564: The labels for $\Delta\theta$ and $\Delta S$ should be next to the colorbar, not on the vertical axes. The "decade" is 1999 to 2010, but if the authors are trying to say "2010 minus 1999", then they should use the word "minus" rather than a hyphen (or a – rather than a - ). The figure caption would be easier for the reader to understand if it began: "Vertical sections of differences in (a) potential temperature and (c) salinity for the decade 1999 to 2010." The figure caption also states that station numbers and fronts are shown in gray and in black, but only black station numbers are showing. It

would be enough to just include black station numbers, and refer readers to Figure 1 for the ALBATROSS station numbers.

Throughout the text: The original Naveira Garabato et al. (2003) paper does not have a hyphen between surnames, yet when cited in this manuscript that paper is given with a hyphen. Sometimes the authors abbreviate decibars as "dbar" and sometimes as "db". Using "dbar" throughout would be consistent with other oceanographic literature, e.g., Naveira Garabato et al. (2003).

---

## Author Response (AR1)

**Review of the paper by M. Dolores Pérez-Hernández, Alonso Hernández-Guerra, Isis Comas-Rodríguez, Verónica M. Benítez-Barrios, Eugenio Fraile-Nuez, Josep L. Pelegrí, and Alberto C. Naveira-Garabato**

**"Properties and mass transport differences across the Falkland Plateau between 1999 and 2010"**

**The authors present a very detailed analysis of water masses in the region of the Falkland Plateau and their comparison between 1999 and 2010 on the basis of two hydrographic sections in the region. However, the main goal of their study is comparison between the properties and mass transport across the Falkland Plateau. The analysis of the differences and causes is not sufficient for immediate publication.**

**This region is the location of the beginning of the Falkland Current but the authors just mention this fact as not very important.**

*We thank the reviewer for his/her comments. In lines 45-48 specify how the ACC contributes to the Falkland current with 60-70 Sv.*

**The cruises were made in different seasons of the year: the austral fall and summer (April and February). The changes in the water properties and dynamics between these two seasons can occur not only in the surface layer as the authors report. It is well known that the seasonal changes in the Falkland Current are very strong. The changes in the geostrophic component of the Falkland Current may reach the depths of 2000 m. Since the changes in the Falkland Current exist, similar changes may occur in the region where the current starts. Some analysis of what had happened between 1999 and 2010 is needed. The authors cite a publication by BÅSning et al. [2008], but this paper analyzes the changes in the ACC caused by decadal changes in the wind field.**

*We agree that it would be interesting to know the seasonal contribution, but in trying to infer this, we did not find any other oceanographic cruises occupying the region, either enough Argo floats (the program started in 2004) as can be seen in the figure below from the Coriolis data centre. This highlights the importance of publishing this study. It provides with a comparison between the only two high-resolution data available in the area. This important point has been clarified in the introduction and discussion of the revised*

*manuscript.*

[Figure]

**I would appreciate an analysis of the AVISO data in the region and variations in the geostrophic currents during the study period (or at least in the period when the AVISO data are available and reliable). The analysis of the seasonal changes in the geostrophic currents is important. Then, this analysis should be linked with the observations performed with an interval of 11 years. The changes that occur over a period of 11 years and the seasonal changes should be separated. The changes in mass transport even in the subsurface layers can be associated with the seasonal changes in winds. I am sure that some CTD data from the stations occupied in the region in the years between 1999 and 2010 can be found in the databases and added to the analysis. The main drawback of the analysis presented in this manuscript is complete absence of any data related to the period of 11 years.**

*We agree in that we should provide with a quantification of the seasonal geostrophic changes between both cruises. We have included an analysis of the AVISO derived geostrophic transport in our section 3.*5 Relative geostrophic transport changes *and in the* discussion *of the revised manuscript. It consists of the following figure and text.*

*In 3.5 Relative geostrophic transport changes*

[Figure]

*Fig. 10. East to west accumulated relative geostrophic mass transport from AVISO averaged from 1993 to 2016 together with their standard deviations: February (black solid line) and April (grey solid line). For this calculation, the depth of 50 m has been considered to compute the mass transport that corresponds to SASW. Dashed lines are the east to west accumulated relative geostrophic mass transports shown in Figure 7 for the SASW for the MOC Austral (black) and ALBATROS (grey) carried out in February and April, respectively.*

*"To put all the estimated transports in context, the monthly 1993-2016 averaged geostrophic velocities from AVISO are interpolated to the station pairs of both cruises and integrated by using the stations distance and the average depth of the SASW stratum (50 m). This is shown in Figure 10, where the 1993-2016 average of all Februaries (Aprils) is contrasted with the estimated relative transport of the MOC Austral (ALBATROS) upper stratum. It is seen that there is no climatological significant difference between the estimations of both months. Hence, the positions and transports (expressed as mean ± standard deviation) of the SAF and PF in the AVISO derived transports are 2.5±0.5 Sv at the longitudinal range 52.97-56.96ºW and 1.1±0.7 Sv at 47.49-51.34ºW, respectively. The SAF AVISO estimated transport is approximately the average between the SASW*

*transports of the ALBATROSS (3.2 Sv) and MOC Austral (1.9 Sv). The PF observed in the AVISO data covers a wider range of longitudes than the ones of the hydrographic surveys. Its transport is slightly smaller than for the ALBATROSS cruise (2.0 Sv) and non-significantly different from the MOC Austral (1.6 Sv) at the SASW stratum (Table 1)."*

**In the discussion:**

*"The AVISO climatological seasonal transport average is non-significantly different between February and April. Thus, the observed changes in transport are due to interannual variability."*

**I can recommend this manuscript for publication only if such analysis would be added. Major revision is currently needed.**

**A few minor remarks. Do not use different notations in one figure. Red and black dots in Fig. 1a and black and gray dots in Fig. 1b.**

*Yes, in the figures when we compared both years we have chosen to use black (grey) for the MOC Austral (ALBATROSS) cruise (Figures, 1b, 7 and 9), but for Figure 1a we have changed the color for ALBATROSS cruise to red, to be able to discern in between both cruise stations.*

**Is Fig 1b is just a copy of the figure from another text?**

*It is not, we have the used the novel temperatures and salinities from the MOC-Austral cruise and combined them with the ones from 1999 to see the difference between both years in a unique plot.*

**What are the units of color scale in Figs. 3 and 4? I recommend changing color in one of the color scales not to use tones of red and blue in both color scales with different units. Do not use the same colors for different properties.**

*For Figure 3 the units are cm/s and we agree that they should be shown in the figure or caption. We apologize for the inconvenience; this error has been fixed in the caption of Fig.3 of the revised manuscript. In contrast, Figure 4 clearly shows its units on the y-axis.*
*We have followed your advice on using different color scales for each variable on the revised manuscript being: (1) velocities, red to blue, (2) potential temperature anomalies, orange to blue and (3) salinity anomalies, purple to green.*

**General comments: This manuscript presents new data from 27 full-depth CTD stations across the Falkland Plateau in February/March 2010. The CTD stations were deliberately located at the same latitude and longitude positions of the April 1999 ALBATROSS cruise, to allow comparisons between the two data sets. The authors compare locations of fronts, water masses, and geostrophic volume transports (net, and for each water mass). This area is not an easy area to collect oceanographic data, because it is remote and has the Sub-Antarctic Front and Polar Front passing through it. The data presented are therefore important, and deserving of publication. Although well-written, the paper is lacking in some explanations, as detailed below in the specific comments. With revisions, this manuscript would be a useful addition to the literature.**

*We thank the reviewer for his/her careful reading of the manuscript and his/her comments.*

**Specific comments: As noted above, although the data are important and deserving of publication, there are some revisions needed.**

**Line 1: The title "Properties and mass transport differences across the Falkland Plateau between 1999 and 2010" does not express what the importance of the paper is. The values calculated are "volume transport" (units of metres cubed per second), not "mass transport" (units of kilograms per second). "Properties" is too vague – which proper- ties? A title that summarises the key message or focus of the paper is needed.**

*We agree that the paper can use some clarification in stating which kind of transport units are used throughout the text. In line 32 we have described the relation between the units of mass and volume transport and specified that transport will be expressed in Sv. In addition, later in line 86 we have clarified now that the study uses mass transport.*

*We have also changed the title to include the words "fronts" and "water masses".*

**Lines 13-28: The abstract summarises some of the main points, but does not tell the reader why they should read the full paper. At line 23, the term to "relative mass transport" is distracting, because what the authors have really looked at is "volume transport". The changes in width and volume transport of the SAF and**

**PF from 1999 to 2010 are listed, but it is not clear why these have occurred or why the changes might be important. The abstract should be rewritten to draw the reader into the full paper.**

*Please see our previous comment relate with the transport units. A sentence has been added in line 25 to explain part of the changes observed in the fronts.*

**Lines 34-36: The authors set the scene at the opening lines of their paper by citing Orsi et al. (1995) and Cunningham et al. (2003) in regards to ACC transport being between 100 and 150 Sv. However, there are a number of very recent papers that indicate that ACC transport might be much higher than this (Donohue et al. (2016) and references therein (at page 11,766)). It would be worth the authors including discussion of the implications (if there are any) of those papers to the research presented in this manuscript.**

*We agree that Donohue et al. (2016) is clearly a relevant paper as it describes a higher transport value for the ACC. Hence we have modified our ACC transport range on the introduction to add this new citation (Line 33).*

**Lines 87-88: With respect to the 2010 data, the authors state: "Relative geostrophic velocities are estimated using the sea bottom as level of no-motion." However, Naveira Garabato et al. (2003) state for the 1999 data: "At each station pair, the reference velocity is initialized by performing a least squares fit of the geostrophic shear profile to the average detided water-track LADCP profile", which (if I am reading this correctly) means the 1999 data used a different method to obtain geostrophic velocities (by adjusting them to LADCP measurements). Naveira Garabato et al. (2003) in their section 3 then go on to discuss the issues with level of no motion assumptions. The authors of the present manuscript therefore need to explain why they used the level of no motion approach for the 2010 data, and whether they re-analysed the 1999 data using the same assumption or if they are using the 1999 data as processed by Naveira Garabato et al. (2003).**

*In the last paragraph of section 3, Naveira Garabato et al. (2003) explores the importance of having in situ observations. To do this they use the deepest common level between stations stating that is "a common practice in invers box models of the Southern Ocean". In the cruise carried out in 2010, LADCP data was not available. Thus transport is estimated for both surveys using the bottom as reference level in order to compare between both cruises. This has been clarified in lines 86-87.*

**Table 1 (lines 425 onwards): The first three columns are ALBATROSS (1999) data, but the values in the table do not seem to match those given in Figure 14 of Naveira Garabato et al. (2003). For example, Table 1 states UCDW volume**

**transport was 12.4 Sv, whereas Naveira Garabato give that volume transport as 23.5 ± 4.5 Sv. From reading the manuscript, I thought the same data was being used, so if a new analysis has been done, or there is some other reason for the values being different, then this should be explained.**

*Figure 14 from Naveira Garabato et al. (2003) reflects their final absolute geostrophic transports. This has been estimated by adjusting the relative geostrophic transports with in situ velocities from a LADCP and with an inverse box model. As stated in the previous response to the reviewer, LADCP data were not available for the 2010 survey and therefore we cannot estimate absolute geostrophic transport. Hence, in order to compare both cruises we have estimated relative geostrophic transport for the 1999 and the 2010 cruise data.*

**Lines 265-335 (Discussion and conclusions): The discussions and conclusions do not really convey why the research was important. The main conclusions as presented in this section seem to be: (1) the observed changes in surface waters are because the measurements in 1999 (April=austral fall) were performed in a different season to 2010 (February=austral summer); (2) the intermediate waters underwent changes that are attributable to wind changes at their regions of formation; (3) the salinity and temperature do not change much for the CDW layers.**

*At the end of the discussion we have added a few sentences on the importance of the observed changes in the overall circulation to highlight the importance of this study (lines 363-369).*

**However, the conclusions above do not really connect with some of the statements in the "Results" section, e.g., lines 163-169, where it states "In between stations 9-12 where the SAF stands, the UCDW exhibits a remarkable increase in salinity." I also found it surprising that the authors did not directly mention the 1998-1999 ENSO and SAM events that Naveira Garabato et al. (2009) say were so important for the AAIW properties, and the implications therefore of comparing the ALBATROSS results with the 2010 ones. There needs to be some overall important conclusions presented from the research.**

*Lines 166-167 report an increase in salinity in the UCDW localized only in the SAF. This increase is compensated with the negative anomalies existing in the stratum, providing an average non-significantly different from zero. We have clarified this for the reader in the next sentence (lines 166-167).*

*The SAM is now included in the discussion. The reviewer has made an important point in emphasizing the importance of this mode for the winds and the circulation.*

**Technical corrections: Lines 3-5: It is completely up to the authors how they wish to have their names presented, but I have never seen Professor Naveira Garabato's paternal and maternal surnames hyphenated before.**

*Thanks for this observation, this has been corrected in line 5.*

**Line 56: "data" is plural, so this line should read "data from this cruise have been.**

*Agreed, this has been changed in the revised manuscript.*

**. ." Line 57: Delete "with"**

*Done*

**Line 59: "confronted" is the wrong word to use here.**

*The word has been changed to "compared"*

**Line 66: "realizations" is the wrong word to use here**

*It has been changed to "surveys"*

**Lines 67-73: This paragraph basically says that the paper is organized with data and methods first, then results, then discussions and conclusions. Almost all scientific papers are organised that way, so this paragraph could be deleted. A more useful thing to have in its place would be a sentence that starts with "In this paper,. . .", followed by the most important thing(s) that the paper shows/examines.**

*We would like to keep this paragraph. Although it does not provide with important information it presents the outline of the manuscript, so the reader can decide what parts are more relevant for his/her purpose. On the other hand, the previous paragraph (lines 59-65) already states the important subjects that the paper examines.*

**Line 82: "accurary" should be "accuracy" (presumably)**

*Thanks, fixed.*

**Line 83: An accuracy of 0.001 C would mean that the SBE911+ CTD on the ship was new from the factory or had been recently calibrated (or post-cruise factory calibrated), because the specifications of the SBE911+ CTD say this is the initial accuracy, with a drift of "0.0002 C per month" (http://www.seabird.com/sbe911plus-ctd, under specifications tab). This may not be important for the level of accuracy required for the measurements presented in this manuscript, but more details should be given.**

*The CTD was sent to SeaBird for calibration before the cruise.*

**Line 86: I have no issues with the Practical Salinity Scale (rather than Absolute Salinity from TEOS-10) being used in this research, especially because the authors are making comparisons with earlier published work that was published using PSS. However, the reference for PSS should be cited at the end of this sentence.**

*As the reviewer says we have used the Practical Salinity Scale and this is specified in lines 84-85 of the manuscript to avoid confusion.*

**Lines 89-90: ". . .modifications attending to the present water masses" – it is not clear to me what this means.**

*It means what it stated in the next sentence, that WSDW is not present in 2010 and therefore LCDW is considered as having densities higher than 27.90 kg m$^{-3}$, which differs from Naveira Garabato (2003). To avoid misunderstanding we have deleted the part of the sentence that said "modifications attending to the present water masses"*

**Line 193: "doesn't" is too informal for a scientific paper, please change to "does not".**

*Changed*

**Lines 235-236 and 238-239: At lines 235-236, the authors state ". . .the SAF-associated jet is displaced westward and weakens 14.7 Sv as compared with the ALBATROSS observations", then at lines 238-239 the authors state "The relative net transport is 9.2 Sv greater during the ALBATROSS cruise as an outcome of a more intense SAF." It would be easier for the reader to follow this line of argument if the comparisons were consistent, so something like this: ". . .the SAF-associated jet in 2010 was displaced westward and was weaker by 14.7 Sv compared with the ALBATROSS observations", then "The relative net transport is 9.2 Sv less during 2010 compared to during the ALBATROSS cruise as an outcome of a weaker SAF in 2010." How 14.7 Sv less volume transport of the SAF leads to 9.2 Sv reduction in net transport would also be worth explaining , that is why volume transport changes are not directly additive.**

*We have changed the first two paragraphs of section 3.5 following this comment from the reviewer.*

**Lines 255-256: "The total transports. . ." – the authors have already stated these results at lines 239-241, so this sentence could be removed.**

*As the reviewer highlights there was some repetition on those lines. Therefore the sentence about the net transport in the first paragraph has been moved down and the*

*second paragraph has been modified (lines 248-252).*

**Figure 1 at lines 491-496: It looks like the 2010 stations were at the same locations as the ALBATROSS stations on the scale of this map, so rather than saying "red dots" and "black dots", perhaps just "red station numbers" and "black station numbers" would be sufficient. The figure caption should also say "Hydrographic stations across the Falkland Plateau. . .", because ALBATROSS included other stations not shown on this map.**

*Thanks for the suggestions. Changes have been made in the caption of figure 1.*

**Figure 4 at lines 522-564: The labels for ✓ and S should be next to the colorbar, not on the vertical axes. The "decade" is 1999 to 2010, but if the authors are trying to say "2010 minus 1999", then they should use the word "minus" rather than a hyphen (or a – rather than a - ). The figure caption would be easier for the reader to understand if it began: "Vertical sections of differences in (a) potential temperature and (c) salinity for the decade 1999 to 2010." The figure caption also states that station numbers and fronts are shown in gray and in black, but only black station numbers are showing. It would be enough to just include black station numbers, and refer readers to Figure 1 for the ALBATROSS station numbers.**

*We agree that the y-axis is not the best place to put the title and units of the figure. Furthermore, we have considered that the figure needed labels. Thus, labels have been added together with the title of the figure and the caption has been modified following the reviewer's suggestions.*

**Throughout the text: The original Naveira Garabato et al. (2003) paper does not have a hyphen between surnames, yet when cited in this manuscript that paper is given with a hyphen.**

*This has been fixed in the revised manuscript.*

**Sometimes the authors abbreviate decibars as "dbar" and sometimes as "db". Using "dbar" throughout would be consistent with other oceanographic literature, e.g., Naveira Garabato et al. (2003).**

*We apologize for this mistake. "dbar" is the correct unit expression. This has been corrected throughout the text.*

---

## Author Response (AR2)

*On behalf of my coauthors I would like to thanks both reviewers for their second review of the manuscript. Their comments and suggestions have considerably improved the manuscript. Reviewer #1 suggests publishing as it is while reviewer #2 suggests some technical corrections. Thus, we will carefully respond to each comment of reviewer #2 beneath.*

**Anonymous Referee #2**

**The authors have taken on board many of the comments in my original review, but not all of them. Perhaps my explanations were not clear enough. I have endeavoured to give clear explanations in my comments below. As with the earlier version, with revisions, I believe that this manuscript would be a useful addition to the literature.**

**In my original review, I noted that: "The title "Properties and mass transport differences across the Falkland Plateau between 1999 and 2010" does not express what the importance of the paper is. The values calculated are "volume transport" (units of metres cubed per second), not "mass transport" (units of kilograms per second). "Properties" is too vague – which properties? A title that summarises the key message or focus of the paper is needed."**
**The authors' response was: "We agree that the paper can use some clarification in stating which kind of transport units are used throughout the text. In line 32 we have described the relation between the units of mass and volume transport and specified that transport will be expressed in Sv. In addition, later in line 86 we have clarified now that the study uses mass transport. We have also changed the title to include the words "fronts" and "water masses"."**
**The new title is "Differences between 1999 and 2010 across the Falkland Plateau: Fronts and water masses", and this is an improvement. The authors say that they have "specified that the transport will be expressed in Sv", yet they persist in referring to the transport throughout as "mass transport", rather than "volume transport". "Volume transport" is the normal phrase used in oceanography for the property measured in Sv, and I do not understand why the authors insist on calling this "mass transport".**

*On one hand, the title has changed following the suggestion of this reviewer of avoiding the word "properties" on the title and we agree that it has improved the quality of the paper. On the other hand, the unit Sv can be used for volume or mass transport as can be seen in the following equivalence: $1\ Sv = 10^6\ m^3\ s^{-1} \approx 10^9\ kg\ s^{-1}$. A numerous amount of articles in physical oceanography use Sv for mass transport instead of volume transport. We will like to keep on using it. To alert the reader the word "mass" was added in most of the cases where we refer to transport. Here I'm showing some the latest works using Sv as mass transport in physical oceanography:*

- Döös K., J. Kjellsson, Z. Jan, F. Laliberté, L. Brodeau, A. Aldama Campino

(2017). The Coupled Ocean-Atmosphere Hydrothermohaline Circulation. *Journal of Climate.* Vol 30, pp 631-647. Doi: http://dx.doi.org/10.1175/JCLI-D-15-0759.1

- Evans G.R., E.L. McDonagh, B.A. King, H.L.Bryden, D.C. E. Bakker, P.J., Brown, U. Schuster, K.G. Speer, S.M. A. C. van Heuven (2017). South Atlantic interbasin exchanges of mass, heat, salt and anthropogenic carbon. *Progress in Oceanography*, Vol 151, Pages 62-82. Doi: https://doi.org/10.1016/j.pocean.2016.11.005

- Vélez-Belchí, P., M. D. Pérez-Hernández, M. Casanova-Masjoan, L. Cana, and A. Hernández-Guerra (2017), On the seasonal variability of the Canary Current and the Atlantic Meridional Overturning Circulation, *J. Geophys. Res. Oceans*, 122, doi:10.1002/2017JC012774.

- Hernández-Guerra A., E. Espino-Falcón, P. Vélez-Belchí, M.D. Pérez Hernández, A. Martínez-Marrero, L. Cana (2017). Recirculation of the Canary Current in fall. *Journal of Marine Systems*. doi: https://doi.org/10.1016/j.jmarsys.2017.04.002

- Pontes, G. M., A. Sen Gupta, A. Taschetto (2016). Projected changes to South Atlantic boundary currents and confluence regions in the CMIP5 models: the role of wind and deep ocean changes. *Environmental research Letters II*. Vol 11, 9.

**In my original review, I noted for what are now lines 66-72: "This paragraph basically says that the paper is organized with data and methods first, then results, then discussions and conclusions. Almost all scientific papers are organised that way, so this paragraph could be deleted. A more useful thing to have in its place would be a sentence that starts with "In this paper,. . .", followed by the most important thing(s) that the paper shows/examines." The authors' response was: "We would like to keep this paragraph. Although it does not provide with important information it presents the outline of the manuscript, so the reader can decide what parts are more relevant for his/her purpose. On the other hand, the previous paragraph (lines 59-65) already states the important subjects that the paper examines." I don't know any readers who look for a paragraph at the end of the introduction to work out which parts are relevant to them. Most people with whom I have discussed "how do you read/review a paper?" say that they read the abstract first, then go through all the figures, then read the conclusion to see if it matches what they think the figures show, then look at the section headings in sequence, then read the text. I think the authors could strengthen the sentence at lines 62-65 to say what was achieved (currently it is stated as an objective), and**

**finish the introduction there.**

*In regard to this comment, the final paragraph of the introduction now reads as follows (lines 59-68):*

> "In this study, the water masses, relative geostrophic velocities and transports across an almost zonal hydrographic section carried out in 2010 along the Falkland Plateau are evaluated. These data, together with the ALBATROSS cruise, are the only high-resolution hydrographic data available on the region. Thus, results from the 2010 cruise are compared with those obtained from the 1999 cruise in the same area [Naveira Garabato *et al.*, 2003], with the objective of assessing possible relative transport and water mass differences between the two surveys. For changes in the relative transport the position of the fronts and the season in which each cruise took place will be considered. Changes in water mater masses, are decomposed into changes in the θ/S isobaric surfaces and results from the Bindoff and McDougall [1994] model."

**Line 81: In my original review, I noted that: "An accuracy of 0.001ºC would mean that the SBE911+ CTD on the ship was new from the factory or had been recently calibrated (or post-cruise factory calibrated), because the specifications of the SBE911+ CTD say this is the initial accuracy, with a drift of "0.0002ºC per month" (http://www.seabird.com/sbe911plus-ctd, under specifications tab). This may not be important for the level of accuracy required for the measurements presented in this manuscript, but more details should be given." The authors' response was: "The CTD was sent to SeaBird for calibration before the cruise." However, the authors' have not added this information to the manuscript. I believe it is an important detail, as not all CTDs are sent for pre- and post- cruise calibrations. At line 81, after the words "...and temperature sensors.", the authors should add the sentence "The CTD was sent to SeaBird for calibration before the cruise."**

*We agree, this has been added in line 77 where the reviewer suggests.*

**Lines 84-85: The authors have misunderstood my suggestion, and they have not made any changes to the manuscript in this respect. In my original review, I stated: "...the reference for PSS should be cited at the end of this sentence." The authors have responded with: "As the reviewer says we have used the Practical Salinity Scale and this is specified in lines 84-85 of the manuscript to avoid confusion.", but they have not made any changes to the manuscript. By "reference", I meant that the authors should cite the paper/report that the PSS derives from: Unesco. 1981a. The Practical Salinity Scale 1978 and the International Equation of State of Seawater 1980. Techn. Pap. mar. sci, 36: 25 pp.**

*Following this suggestion the citation has been added to the text and the reference list.*

**Section 3.5: In my original review, I suggested rewording parts of section 3.5 to say: "The relative net transport is 9.2 Sv less during 2010 compared to during the ALBATROSS cruise as an outcome of a weaker SAF in 2010." I also suggested "How 14.7 Sv less volume transport of the SAF leads to 9.2 Sv reduction in net transport would also be worth explaining, that is why volume transport changes are not directly additive." The authors state in their response: "We have changed the first two paragraphs of section 3.5 following this comment from the reviewer." The authors have changed the start of the second paragraph to read as follows: "The relative net transport during the MOC-Austral cruise is 9.2 Sv weaker than in the ALBATROSS cruise as an outcome of a more intense SAF. SASW, AAIW/AASW and UCDW present lower values in 2010 than in 1999, being the surface and intermediate stratums the ones with the highest decadal transport differences (Figure 7 and table 1)." The version of the first sentence that I suggested, "...weaker SAF in 2010", and the authors' new version, "more intense SAF" seem to be contradictory. Figure 3 and Figure 7 show that the SAF was weaker in 2010, but the way the authors have put "more intense SAF" with no qualifiers makes the reader infer that the SAF was more intense in 2010 (MOC-Austral cruise). I suggest that the words "more intense SAF" be followed by "in 1999" for clarity.**

*"In 1999" has been added at the end of this paragraph following this suggestion.*

**Figure 1: The authors have mostly followed my suggestions for changes, except that they have omitted the word "black" for the 2010 station numbers. I suggest this caption is modified to read "...MOC-Austral (2010, black station numbers)..." I agree with the other reviewer that it would be better to have consistent coloring of stations between Figure 1(a) and Figure 1(b) (and later figures). Changing the red station points/numbers in Figure 1(a) to grey would be the easiest way to do that.**

*The word "black" has been added.*

**Figure 10: This figure was added in response to a suggestion from the other reviewer. The figure shows accumulated relative geostrophic mass transport from AVISO 1993-2016 for all months of February and April, as well as ALBATROSS (April 1999) and MOC-Austral (February 2010) data. The AVISO February and**

**April data look like a smoothed version of the ALBATROSS data, and are very different from the MOC-Austral data. In the discussion section, more discussion on this would be interesting. Also, what was Figure 10 in the earlier version of the manuscript (maps of wind stress) is now Figure 11, but that figure is still referred to as Figure 10(a,b,c), which needs to be corrected.**

*Figure 10 reflects how the observed differences between the 1999 and 2010 transport estimates are due to interannual variability and not to seasonal changes. This is explained I lines 338 to 340.*

*The naming of the figures has been checked.*

---

## Author Response (AR3)

**Topic Editor Decision: Publish subject to technical corrections**
(07 Jun 2017) by Mario Hoppema
Comments to the Author:
Dear Dr. Pérez-Hernández and co-authors,

Your manuscript is now accepted for publication in Ocean Science. There are only some few technical corrections, which I request you to take into account.

With best wishes
Mario Hoppema

Non-public comments to the Author:
List of minor comments

L103 … 10 and 20 February 2010 … (date format)

*Changed*

Section 3. Results:
I think when reporting results the common tense used is the past tense, and not the present tense as this is done here.

*If you think that this will improve the manuscript I will change it. However, the whole paper mainly is written in present tense, thus if I change one section it will lose its consistency. Please let me know.*

L181 Temperature and salinity are not observed in the Figure, but in reality. I suggest to change this to: … per decade is observed (Figure 5a and b, respectively).

*Changed accordingly.*

L181-182 This increase can also be seen in Figure 5c. (or alternatively to "seen": discerned)

*Changed*

L187 delete: in

*Changed*

L217 delete: in

*Changed*

L218 slight

*Changed*

L224 suffered? Can be deleted, right?

*Yes, erased*

L225 The result in this stratum …

*Changed*

L250-251 "The LDCW stratum does not registers the SAF due to the bathymetry." This sentence is not clear to me. Please improve.

*The sentence has been changed to the following: "The SAF does not have a contribution from the LDCW stratum due to the shallow bathymetry"*

L294 & 297 larger (instead of: higher)?

*Changed*

Please go through the references and check the Ocean Science format. Add: and before the last author, do not use capitals in the title (except for names), year at the end etc.

*The references have been changed according to the Copernicus Publications Reference Types document available on line.*

Figure 1: the lower part of the figure has been cut off.

*Good point, this has been modified too.*